# OpenReview forum: "Can Reasoning Models Obfuscate Reasoning? Stress-Testing Chain-of-Thought Monitorability"
_ICLR.cc/2026/Conference — ICLR 2026 Conference Desk Rejected Submission_

### Official Review · Reviewer_gUWE · 2025-10-30

**Soundness:** 2
**Presentation:** 3
**Contribution:** 2
**Rating:** 2
**Confidence:** 4

**Summary:**

The paper measures whether models can obfuscate their CoT against monitors under varying degrees of obfuscation pressure (e.g. being told the CoT is monitored). The main evaluation is performed in an environment where LLMs have to complete legitimate main- and secret side tasks. The paper shows that strong obfuscation pressure allows some models to evade monitors. Additionally, the paper notices that external CoT (CoT within output) is obfuscated more successfully, though the effect might be due to instruction following differences.

Mandatory disclosure of LLM usage by the reviewer: LLMs were utilized to structure the review content into prose paragraphs and numbered lists.

**Strengths:**

1. CoT monitoring is an important area of AI safety, and CoT obfuscation a timely issue.

2. The paper considers different levels of obfuscation pressure (small clues to detailed obfuscation instructions).

3. The SHADE arena tasks resemble realistic risks (one could imagine the money task happening in reality via a prompt injection on an AI agent email assistant).

**Weaknesses:**

1. The most important weakness is that prompted obfuscation "pressure" may not be analogous to misalignment and scheming which the paper discusses in the introduction. As the discussion section acknowledges, more realistic settings (e.g. fine-tuning, maybe reinforcement learning against the CoT) could produce vastly different outcomes.

2. The paper's results on internal vs external CoT seem to be largely explained by the observation that internal CoT follows reasoning instructions less reliably (section 5.2), I am confused why point (iii) is still stated as a contribution in the abstract.

3. It's not clear how robust the results are to variations in the obfuscation pressure prompts. The appendix mentions different variants of the prompts, but it is not clear from the paper how much the results depend on the particular choices of prompts. This is especially important because the main experiment tests just two obfuscation levels ("low"/"high"). The paper would be improved by studying how much different choices of prompts change the results.

4. Finally, the result that CoT-only monitoring is more accurate than full (CoT+output) monitoring (lines 368 - 370) is surprising and makes me suspicious of the statistical power of the results. The paper could be improved by establishing more clearly that the results are robust.

5. (Minor) The paper uses GPT5-mini as grader, it would be nice to know whether the error rate of the grader significantly impacts the results (or is it basically perfect?).

Nitpicks (not affecting my score):

* In the sentence "some models successfully complete adversarial tasks while evading detection, though this effect is model-dependent", "some models" and "model-dependent" is maybe repetitive.

* In section 3.2 it is not clear why the dose response experiments are conducted; this becomes clear only later in the paper (computational limitation to "low"/"high" settings). Explaining this earlier would improve the flow.

**Questions:**

See weaknesses.

---

> ### Author Response · Authors · 2025-11-21
>
> We thank the reviewer for their thorough assessment and constructive feedback. We address the main concerns below.
>
> ## 1. Prompted obfuscation "pressure" may not be analogous to misalignment/scheming
>
> We thank the reviewer for raising this important distinction. This was indeed a central consideration in our experimental design. We selected prompt-based obfuscation for two key reasons:
>
> **First, we focus on eliciting current models’ capabilities for CoT obfuscation**, then assessing their impact on monitoring. For this reason, stronger elicitations that update model parameters are not necessarily more realistic. This approach allows us to assess where current frontier models stand in terms of CoT manipulation capabilities.
>
> **Second, our approach aligns with established methodologies in the field.** Recent work has demonstrated that concerning model behaviors can be elicited through prompting alone:
> Jones et al. (2025) successfully elicited power-seeking, self-preservation, and self-exfiltration behaviors using only prompting
> Sharma et al. (2023) studied sycophancy using prompt-based methods
> Greenblatt et al. (2024) demonstrated alignment faking through prompting, though they found fine-tuning and RL increased occurrence rates
>
> These studies establish prompting as a valid methodology for studying potentially dangerous capabilities.
>
> We acknowledge that fine-tuning or RL could produce different outcomes, and we view this as important future work building on our capability assessment.
>
> ## 2. Benign instruction following VS Prompted CoT obfuscation
>
> We agree that the two results are related. However, **we believe there are important distinctions due to the adversarial nature of CoT obfuscation:**
>
> 1. **Different task domains**:Benign instructions test presentation-level compliance (formatting, language). Obfuscation pressure targets concealment of hidden objectives—a qualitatively different cognitive demand involving stealth, deception, and monitor-awareness.
>
> 2. **Divergent patterns across models**: Empirically, these two domains exhibit divergent patterns. Models like GPT-5, Magistral Medium and DeepSeek R1 achieve strong benign reasoning instruction compliance (Figure 7) yet show low obfuscation ability (Figure 2). Claude Opus 4.1, for example, has a much smaller internal vs external CoT gap for benign instruction following than for obfuscation. This asymmetry cannot be inferred from Section 5 alone (Section 4.2.1).
> 3. **Adversarial obfuscation also impacts side-task execution** (Section 4.2.2), where internal and external CoT diverge materially—again in ways not predictable from benign instruction following.
>
> For these reasons, we keep contribution (iii) as a standalone result: the internal–external CoT gap specific to adversarial pressure.
>
> ## 3. Robustness of the obfuscation pressure prompts
>
> We agree that we do not systematically explore all strong prompts in SHADE-Arena. So **we updated the paper to clarify the scope of our robustness claims** by (i) adding a short explanation of the two-stage design in the Methods section: Stage 1: Systematic Exploration (Toy Tasks) & Stage 2: Representative Stress Test in realistic settings, (ii) explicitly noting that the SHADE-Arena results are based on a single high-pressure prompt chosen using our cue taxonomy, and (iii) adding a brief summary in the main text clarifying that the key qualitative patterns in the toy experiments are stable across many high-pressure stacks.
>
> ## 4. Monitor results: statistical power and error rate
> We appreciate the reviewer’s close attention to this result. We agree this result is counterintuitive, and for that reason **we conducted additional experiments and extensive analyses to confirm that it is statistically robust and explained by monitor capability limitations.**
>
> **Explanation:** Less capable monitors (GPT-4o-mini, GPT-4o) sometimes misinterpret side-task actions as legitimate user requests, treating associated reasoning as benign. This confusion doesn't occur with CoT-only views. GPT-5 largely closes this gap (80.4% full vs. 97.6% CoT-only), confirming this is a capability limitation.
>
> **Statistical robustness:**
> - Non-overlapping 95% Wilson confidence intervals. N=546 without obfuscation, N=186 with where the side task is successful.
> - Pattern replicates across all three SHADE-Arena task pairs
> - CoT vs. full gap narrows with stronger monitors: 70.4pp (GPT-4o-mini) → 56.1pp (GPT-4o) → 17.2pp (GPT-5)
>
> ## 5. Writing improvements
> We have implemented your suggestions:
> - Abstract now reads: "Under strong obfuscation pressure, some models successfully complete adversarial tasks while evading detection".
> - Section 3.2 now contains explicit motivation for dose-response design.
>
>
> **Thank you again for taking the time to review the paper and providing helpful feedback! Do the above actions address your concerns with the paper? If not, what further clarification or modifications could we make to improve it?**

---

> > ### Comment · Reviewer_gUWE · 2025-11-21
> >
> > The author's response clearly addresses weaknesses 1, 2, and 4. Weakness 3 is still present but documented better. This has resolved the majority of my concerns with this submission, and I have updated my review scores accordingly.

---

### Official Review · Reviewer_Jjtg · 2025-10-31

**Soundness:** 3
**Presentation:** 3
**Contribution:** 2
**Rating:** 4
**Confidence:** 4

**Summary:**

This work explores how can reasoning models obfuscate their reasoning traces, an important question for chain-of-thought (CoT) monitorability. The authors perform this task via a series (and combinations) of prompt cues encouraging the language model to obfuscate the CoT. They then evaluate how successful they are on both internal and external CoT traces in SHADE-arena, targeting tasks involving financial transactions, social media recommendations and aggregating school election results while adding hidden side-tasks. The main results highlight that (a) the Claude family of models are more prone to CoT obfuscation when prompted to do so, (b) obfuscation pressure generally reduces the side-task success rate, (c) in the absence of obfuscation cues, CoT monitoring is effective and (d) internal CoT traces are more robust than external ones with respect to the obfuscation cues. The authors finally tie this to how effective are the classes of models used in following instructions, as the obfuscation is encouraged via specific prompt instructions, showing that generally internal CoT follows instructions less reliably, partially explaining the prior findings.

**Strengths:**

- This work provides practical insights on the relevant problem of CoT monitor evasion via adversarial prompting
- Relatively realistic experimental setting, involving multi-turn interactions on SHADE-arena.
- The separate analysis of internal and external CoT was interesting
- The paper was generally well written and easy to follow

**Weaknesses:**

- Some of the results and observations can be seen as direct consequences of the works of (Emmons et al., 2025) and (Baker et al., 2025), limiting the novelty. More specifically, Baker et al., 2025 already demonstrated that training against a monitor can lead to obfuscation (and prompting can be seen as a simple way to "train" a model to obfuscate without changing the model parameters) and Emmons et al., 2025 demonstrated that when the CoT is necessary for the task, it is harder to obfuscate (and thus the results of section 4.2.2 are a direct consequence where the task is replaced with a side-task)
- No actionable recommendations provided in order to mitigate the adversarial prompt effects
- Obfuscation via prompting seems to mostly be seen in the Claude family of models but not as much on, e.g., GPT models and Deepseek,   so the generality of the insights is unclear

**Questions:**

Some comments and questions for the authors

- How is the main task and side-task success rate of the models shown in Figure 2? I believe that grouping the models on Figure 2 by main task performance would better highlight what the models are doing in practice (e.g., is GPT-5 mostly focusing on the main task and successfully solves it whereas it ignores the side-task and thus does not trigger the conditional obfuscation check?)
- On section 4.2.2 the authors claim that these results suggest that obfuscation suppresses secondary behavior more severely than primary task completion; isn't that straightforward, given that the obfuscation pressure is only for the side-task?
- Furthermore, on the same section, the authors mention that a CoT monitor does better than an action or CoT + action monitor. Why is this the case? Couldn't the CoT + action monitor just ignore the action if it is detrimental?
- On the internal vs external CoT discussed on section 5.2; do the authors have specific intuitions into why do the models have higher instruction following rate in the external part than the internal part? At the moment, the authors present just raw results without much discussion about why they manifest.

Overall, based on all of my prior comments, I would say that this work is borderline.

---

> ### Author Response · Authors · 2025-11-21
>
> Thank you for your thorough review and for acknowledging the practical relevance of our work. We answer the questions below.
>
> ## 1. Figure 2: Main vs. side-task success rates
> **Figure 2 shows deliberate obfuscation, not task failure.** The conditional obfuscation metric filters out cases where models ignore the side-task—a model only enters the denominator after successfully completing both tasks. Low bars mean the monitor detected hidden goals despite successful execution, not task failure.
>
> **We added Figure 9 in App. to show both-task success rates (40-80% for most models) vs. conditional obfuscation**, confirming that non-Claude models with near-zero conditional obfuscation are completing side-tasks but getting detected.
>
> ## 2. Side-task degradation under CoT obfuscation
>
> **The magnitude and asymmetry are non-obvious despite directional expectations.** While it's intuitive that obfuscation targeting side-tasks would impair them more than main tasks, the result is less straightforward than it appears because main and side tasks share substantial overlap—they use the same tools, occur in interleaved turns, and require coordinated reasoning.
>
> **Key empirical findings:**
> 1. **Substantial, capability-dependent degradation:** Both the degradation magnitude and residual capability vary by model: For internal CoT, Sonnet 3.7 shows 40pp drops but ends at only 21% success, while Sonnet 4 retains 45% success despite obfuscation (39pp drops). This demonstrates that obfuscation imposes large but not insurmountable costs, with stronger models maintaining dangerous levels of hidden-goal capability.
>
> 2. **Channel-specific asymmetry:** The 13pp difference between internal and external degradation reveals that obfuscation mechanisms interact differently with reasoning channels—a pattern not predictable from the directional expectation alone
>
> 3. **Favorable for defense:** This creates a dual-protection property: models struggle to simultaneously conceal intent *and* complete hidden objectives. Even when CoT monitoring fails, the hidden goal often fails too.
>
> ## 3. Monitor Results: CoT-only outperforming full monitoring (CoT + action)
> Please see response #4 to reviewer gUWE.
>
> ## 4. Actionable Recommendations
> **We added an explicit mitigations to the Discussion. Our findings directly inform several practical mitigations:**
> 1. **Model-level stress testing:** We observe strong model-specific variations: Sonnet 4 retains >94% detection even under heavy obfuscation, while Sonnet 3.7 collapses to <15% (Fig. 5).
> 2. **Prioritize internal CoT monitoring:** Internal reasoning is increasingly harder to obfuscate in newer models (57pp gap for Sonnet 4.5 vs. 8pp for Sonnet 3.7). Avoid optimizing internal CoT for user-facing readability.
> 3. **Pair CoT monitoring with behavioral anomaly detection:** Obfuscation reduces side-task success by 30-45pp (Fig. 4), meaning attacks often fail even when reasoning is hidden.
>  4. **Prompt protection: block or rewrite adversarial cue stacks:** Successful evasion requires stacked cue patterns (Do-Not-Mention + Monitor-Aware + Stealth + structural aids). Systems can detect and rewrite these adversarial prompt signatures.
>
> ## 5. Limited generality of insights
> We respectfully disagree with the limited generality of our insights. **We set out to measure how effectively models can obfuscate their CoT—and found substantial variation across model families.** The fact that Claude models demonstrate stronger obfuscation capabilities than GPT or DeepSeek under the same prompt pressure is itself an interesting and safety-relevant finding.
>
> Additionally, Claude-family findings remain highly valuable:
> - Claude models represent a major deployment share of frontier reasoning systems
> - The Claude family is the only series where both internal and external reasoning traces are accessible, enabling our core comparison
> - Internal CoT is increasingly prevalent (GPT 5, DeepSeek R1, Gemini 3, etc), making channel-specific vulnerabilities increasingly relevant
>
> ## 6. Novelty relative to prior work
> Existing work
> - evaluate only single-channel CoT,
> - do not compute conditional obfuscation (evasion given hidden-goal success), and
> - do not analyze long-horizon, multi-tool agent settings where CoT is operationally central.
>
> Our work extends this line of research in three key dimensions: we
> 1. structure obfuscation pressure into composable cue types and stacks for systematic assessment,
> 2. explicitly measure conditional obfuscation, evasion given successful hidden-task completion, and,
> 3. compare internal vs. external channels side-by-side.
>
> **We updated the Related Work section to further clarify this.**
>
> ------------
>
> **Thank you again for taking the time to review the paper and providing helpful feedback! Do the above actions address your concerns with the paper? If not, what further clarification or modifications could we make to improve it?**

---

> > ### Comment · Reviewer_Jjtg · 2025-11-25
> >
> > I would like to thank the authors for their response. While I appreciate their effort that addresses some of my concerns, unfortunately, it doesn’t convince me enough to update my score. More specifically,
> >
> > 1. Figure 2; the point I wanted to bring across is that this figure hides how often the models ignore the side-task relative to the main task, which is an important piece of information to gauge overall obfuscation / safety behavior. I appreciate the new Figure 9 that the authors share as that adds more information, but it still does not provide information on how much priority do the models give on the primary versus the side task.
> > 2. Could the authors substantiate how much overlap there is between the main / side tasks and relate that to how often the main / side-task fail when there is a push for obfuscation? In line 372 in the manuscript the authors say that CoT obfuscation only affects the side-task and that the performance of the main-task is relatively unchanged. This could highlight that (a) there is not enough overlap or (b) the overlap is not the issue (as obfuscation is a context issue, i.e.., “is this about the side-task” and not a tool overlap issue, i.e., “I have to use the same transfer_money tool for both tasks”). Overall, I am still not convinced that this work provides additional insight on that front relative to what was presented at Emmons et al.
> > 3. The comments provided here resolve my concern. It seems that it is a model capacity issue and that indeed with more capacity the models are able to properly infer whether the action is relevant or not.
> > 4. I appreciate the additional recommendations. This concern is also resolved.
> > 5. I agree that it is a relevant finding but I still question its generality; the fact that, e.g., mostly Claude-family models obfuscate their reasoning via adversarial prompting could also be an artifact of the system prompt / alignment procedure employed in this specific family of models and not necessarily a fundamental issue of reasoning models in general. Furthermore, the claims about internal CoT are a bit confusing; the authors argue that GPT 5, DeepSeek R1 and others use internal CoT but, for GPT 5 at least, they only benchmark with external.
> > 6. I appreciate the elaboration on the delta that this work offers relative to prior work but still, in my opinion, the contribution seems relatively limited and mostly on the internal vs external CoT comparisons (a concern shared by reviewer tyVE as well).

---

### Official Review · Reviewer_tyVE · 2025-11-01

**Soundness:** 3
**Presentation:** 3
**Contribution:** 2
**Rating:** 6
**Confidence:** 4

**Summary:**

This paper investigates whether reasoning models can obfuscate their chain-of-thought (CoT) to evade monitoring while pursuing hidden objectives. It introduces a taxonomy of prompt-level obfuscation cues, compares internal vs. external CoT robustness under escalating pressure, and evaluates models in toy tasks and SHADE-Arena. Key findings: CoT monitoring is highly effective without obfuscation (~96–98% detection) but collapses under strong pressure (<10% for some models); external CoT is more brittle than internal CoT; vulnerabilities are model-dependent.

**Strengths:**

* The paper addresses a highly relevant safety question: Can reasoning models deliberately obfuscate their CoT to evade monitoring? This is timely given the growing reliance on CoT for alignment oversight. The introduction frames the problem well and situates it within recent literature on deception and monitorability.
* Metrics like conditional obfuscation and obfuscated success are well-defined and isolate deliberate concealment from trivial failures.
* Strong evidence that external CoT is more brittle than internal CoT under obfuscation pressure.
* Systematic experimental setup combining toy tasks and realistic multi-turn environments.
* Clear metrics (conditional obfuscation, obfuscated success) that isolate deliberate concealment.

**Weaknesses:**

* Novelty is narrower than claimed:
    * Fragility of CoT monitoring and model-dependent vulnerabilities were shown in prior work (Emmons et al., Arnav et al., Hubinger et al.).
    * The fourth contribution (“pressure-dependent vulnerability”) is not new (Baker et al 2025, OpenAI 2025).
* Most interesting result is the internal vs. external CoT gap; other findings largely confirm known fragility.
* Composable taxonomy contribution is under-supported in the main text:
    * Results are buried in Figures 3, 8, 9 and Appendix B.3; no clear analysis of which cues matter most.
    * No consolidated table or marginal effect discussion.
* Related work section is dense and does not clearly separate prior gaps from this paper's contributions.
* Limited scope: only prompt-level obfuscation; ignores RL or fine-tuning strategies.
* Capability trade-off (stealth vs. side-task success) is noted but not deeply analyzed.

**Questions:**

1. Where are the main results for the composable taxonomy? The contribution is emphasized, but evidence seems buried in figures and appendix without clear discussion of cue importance.
2. How do you justify the ordering of cues in the pressure scale? Is there any theoretical or empirical rationale beyond convenience?
3. Given that prior work already showed CoT fragility and model-dependent vulnerabilities, what is fundamentally new here beyond the internal vs. external comparison?
4. Did you analyze marginal effects of individual cues (e.g., which ones drive most obfuscation)?
5. Could the observed capability drop under obfuscation be leveraged as a detection signal? Any thoughts on this as a defense strategy?

---

> ### Author Response · Authors · 2025-11-21
>
> Thank you for the positive evaluation and detailed questions. We address your main points below.
> ## 1. Composable taxonomy main results & Cue Analysis
> Thank you for raising these important points. **We agree our main results need clearer signposting and justification and have made several updates to the paper.**
>
> **Why we didn't analyze marginal cue effects:** Our primary goal with the taxonomy was to demonstrate that high-pressure stacked cues can induce evasion, rather than to determine the marginal utility of single cues. This design choice was driven by empirical necessity:
> As stated in the paper, Base and single-cue settings failed almost universally, yielding conditional obfuscation strictly below 1% across all models. This suggests that complex, non-linear interactions between multiple cue families are required to achieve meaningful obfuscation.
> Because of this non-linearity, a standard marginal analysis of individual cues could be misleading.
>
> **Main results appear in three places:**
> 1. **Fig. 3 & Fig. 10 (dose-response):**: Figure 3 shows a clear dose-response relationship for susceptible models: stacks with at least 3 cues achieve substantially higher obfuscation rates than low-pressure settings.
> 2. **Fig. 2 & Fig. 9 (aggregate evasion):** Summarizes the success of the taxonomy by plotting the conditional obfuscation achieved (averaged over the top-5 stacks) for each model/channel.
> 3. **Appendix B.3 (Structure of High-Performing Prompts):**Lists the concrete, top-performing stacks (typically 4–6 cue families).
>
> **Justification for Cue Ordering**: Regarding the ordering of cues in the pressure scale, we confirm that the order was fixed before the main experiments for consistency:
> $$\text{Do-Not-Mention} \rightarrow \text{Monitor-Aware} \rightarrow \text{Channel} \rightarrow \text{Start-With} \rightarrow \text{Guard-Lexicon} \rightarrow \text{Stealth} \rightarrow \text{Few-shot}$$
> This order was chosen heuristically to create an intuitive progression: it starts with fundamental prohibitions (Don't Mention) and awareness (Monitor-Aware), and escalates toward more sophisticated incentives and complex demonstrations (Stealth, Few-Shot) in the higher pressure levels. We do not claim this ordering is theoretically optimal, but it is a consistent and transparent method for defining the quantitative pressure axis.
>
>
> **To improve the clarity of the paper, we have made the following revisions:**
> - We have made the non-linear, stacked nature of the required pressure more explicit in the Methods and Results sections.
> - We have included a textual summary in the main paper that discusses the structural patterns observed in the top-performing stacks (Appendix B.3).
> - We have added a small, compact table in the Appendix showing how frequently each cue family appeared among the top-3 stacks for the susceptible models, further illustrating the composition of effective pressure.
> ## 2. Why Prompting Only?
> Please see response #1 to reviewer gUWE.
> ## 3. Novelty
>
> Please see response #6 to reviewer Jjtg.
>
>
> ## 4. More clear related works section
> Thank you for the feedback and **we updated the paper to clearly separate prior gaps from our contributions.** We agree that the original Related Work section compressed multiple threads of literature into a single narrative, which made it harder to distinguish (i) what prior work establishes, (ii) what gaps remain, and (iii) how our paper addresses them.
>
> ## 5. Capability drop under CoT obfuscation as a detection signal
> We thank the reviewer for this suggestion and **we updated section 4.2.2 and the discussion section to reflect this.** We think it's **yes, with caveats:**
> 1. **Current models can hide their intent or achieve the hidden objective, but struggle to do both.** In SHADE-Arena, we show strong obfuscation pressure reduces side-task success by 32–45pp for Claude Sonnet 4 and ~40pp for Sonnet 3.7 (Fig. 4; Fig. 15). This means obfuscation often imposes a substantial cost.
> 2. **But not necessarily good for detecting hidden goals.** In our experiments, the main challenge is that models can selectively suppress CoT for the side task while leaving the primary task almost entirely unaffected—particularly in realistic multi-tool, multi-turn settings like SHADE-Arena (see Fig. 4). So just watching for drops in primary-task performance may give little signal when the model is pursuing a hidden goal.
> 3. **Using model consistency or predictability as a detection signal is promising.** We agree with the reviewer that the broader idea of using consistency-based anomaly detection signals is appealing. A monitor could potentially flag unusual mismatches between a model’s historical competence and its transient behavior.
> -------
> **Thank you again for taking the time to review the paper and providing helpful feedback! Do the above actions address your concerns with the paper? If not, what further clarification or modifications could we make to improve it?**

---

> > ### Comment · Reviewer_tyVE · 2025-11-25
> > **Revised version of the paper missing?**
> >
> > Before I proceed with further comments, I would like to review the latest revision of the manuscript. The authors’ response mentions substantial updates (e.g., clearer taxonomy results, cue frequency table, improved Related Work), but the updated PDF has not been uploaded, and the revisions page displays "No revisions to display". Could the authors please provide the revised version so that these changes can be verified?

---

> > > ### Author Response · Authors · 2025-11-25
> > >
> > > Thank you for responding. We have updated the pdf in submission. This should be viewable via the "PDF" icon in the upper-right corner of this openreview page. Please let us know if this works for you.

---

> > > > ### Comment · Reviewer_tyVE · 2025-11-26
> > > >
> > > > Thank you for uploading the revised manuscript. After reviewing the changes, I acknowledge that some clarifications were added (taxonomy summary, cue ordering rationale, improved related work). However, these do not substantially change my overall assessment. The paper does not feel stronger than before in terms of contribution or depth of analysis.
> > > >
> > > > My core concerns remain:
> > > >
> > > > - Beyond the internal vs external CoT comparison, most findings confirm known fragility and model-dependent vulnerabilities already documented in prior work.
> > > > - Scope is still narrow, limited to prompt-level obfuscation. Stronger elicitation methods such as reinforcement learning and fine-tuning have already been explored in the community (e.g., Hubinger et al., Baker et al.)
> > > > - The novelty claim feels modest relative to the framing in the introduction. Additionally, most of the obfuscation results are concentrated on the Claude family of models. While this is an important deployment-relevant family, the limited generality raises questions about how broadly these findings apply to reasoning models overall.
> > > >
> > > > Given this, the contribution still feels incremental and does not quite meet the bar for acceptance in my view.

---

### Author Response · Authors · 2025-11-21
**Meta Comment**

We thank all reviewers for their constructive feedback. Key changes:

**New Experiments & Analysis:**
1. **Ran more monitor experiments and extended analysis** (gUWE, Jjtg) with expanded statistical testing (Wilson CIs, N=546/186 samples), confirming CoT-only monitoring outperforms full monitoring due to monitor capability limitations.
2. **Claude Sonnet 4.5 experiments** (Jjtg) revealing widening internal-external gaps across model generations (8pp→43pp→57pp)
3. **Comprehensive cue taxonomy analysis** (tyVE) with frequency tables showing which cue combinations drive successful obfuscation across 38 configurations

**Key Clarifications & Paper Updates:**
- **Scope & Methodology** (gUWE, tyVE): Clarified that our work elicits current model capabilities for CoT obfuscation, and assesses the impact on monitor. Not focusing on stronger elicitation that updates model parameters.
- **Prompt Robustness** (gUWE): Explicitly described our two-stage design: Stage 1 explores 38 prompt configurations on toy tasks; Stage 2 stress-tests a representative prompt in SHADE-Arena. Key patterns are stable across high-pressure stacks.
- **Novelty** (Jjtg, tyVE): Strengthened articulation of distinct contributions: (1) prompting alone can elicit CoT obfuscation, which collapses detection from ~97% to <15%; (2) first internal vs. external CoT comparison with large channel gaps (57pp for Sonnet 4.5); (3) realistic multi-tool agent evaluation revealing asymmetric degradation patterns.
- **Actionable Recommendations** (Jjtg): Added actionable recommendations to Discussion.
- **Writing** (All): Revised abstract, restructured Related Work, improved taxonomy presentation with main-text summaries.

---

### Note · Program_Chairs · 2025-11-26
**Submission Desk Rejected by Program Chairs**

Anonymity violation in footnote of revised paper.